# Presumed Onchocerciasis Chorioretinitis Spilling over into North America, Europe and Middle East

**DOI:** 10.3390/diagnostics13243626

**Published:** 2023-12-08

**Authors:** Ahmad Mansour, Linnet Rodriguez, Hana Mansour, Madeleine Yehia, Maurizio Battaglia Parodi

**Affiliations:** 1Retina Service, Department of Ophthalmology, American University of Beirut, Beirut 1107, Lebanon; 2Retina Service, Wills Eye Hospital, Thomas Jefferson Medical Center, Philadelphia, PA 19107, USA; linnetr200@gmail.com (L.R.); hanamansour100@gmail.com (H.M.); 3Retina Service, University of Illinois Chicago, Chicago, IL 60612, USA; madeleineyehia@gmail.com; 4Retina Service, Department of Ophthalmology, Ospedale San Raffaele, University Vita-Salute, 20132 Milan, Italy; maubp@yahoo.it

**Keywords:** blackfly, blindness, chorioretinitis, ivermectin, microfilaria, onchocerciasis, optic atrophy, optic neuritis, river blindness, tropical disease, uveitis

## Abstract

Background: Newer generation ophthalmologists practicing in the developed world are not very familiar with some tropical ocular diseases due to the absence of reports in the ophthalmic literature over the past thirty years. Because of world globalization or due to influx of immigrants from sub-Saharan Africa, exotic retinal diseases are being encountered more often in ophthalmology clinics. Methods: A multicenter case series of chorioretinitis or optic neuritis with obscure etiology that used serial multimodal imaging. Results: Four cases qualified with the diagnosis of presumed ocular onchocerciasis based on their residence near fast rivers in endemic areas, multimodal imaging, long term follow-up showing progressive disease and negative workup for other diseases. Characteristic findings include peripapillary choroiditis with optic neuritis or atrophy, subretinal tracts of the microfilaria, progressive RPE atrophy around heavily pigmented multifocal chorioretinal lesions of varying shapes, subretinal white or crystalline dots, and response to ivermectin. Typical skin findings are often absent in such patients with chorioretinitis rendering the diagnosis more challenging. Conclusions: Familiarity with the myriad ocular findings of onchocerciasis, and a high-degree of suspicion in subjects residing in endemic areas can help in the correct diagnosis and implementation of appropriate therapy. Onchocercal chorioretinitis is a slow, insidious, progressive, and prolonged polymorphous disease.

## 1. Introduction

Onchocerciasis, also called river blindness, is a vector-borne disease under the neglected tropical diseases caused by the tissue nematode *Onchocerca volvulus* (*O. volvulus*). It is highly prevalent in areas where there is an abundance of fast flowing rivers and vegetation. Most of the disease exists in sub-Saharan Africa where it is endemic, with few foci in Yemen and Latin America [1,2]. It is estimated that around 21 million *O. volvulus* infections exist worldwide with 14.6 million having stigmatizing skin disease (severe itching and disfiguring skin conditions), 1.15 million suffering from visual disability, and 10,000 cases experiencing epilepsy (Nodding syndrome or tonic-clonic seizure) [3,4]. Onchocerciasis is transmitted through repeated bites of infected black flies belonging to the genus *Simulium*. Despite the preventive community-directed ivermectin treatment via mass drug administration coverage of around 80%, onchocerciasis transmission remains persistent in many districts [2,5].

Inside the human host, female blackflies release around a thousand microfilariae daily over a life span of 9–13 years. These microfilariae, or embryonic larvae travel to the skin, eyes, and other organs. The microfilariae that are consumed by a female blackfly after biting an infected person grow inside the blackfly and are subsequently passed to the next human host during consecutive bites. Microfilariae measure 300 µm in length and 8 µm in cross-section with a one-year-lifespan. Inside blood vessels, the filariae hardly ever trigger an immunological reaction. Inflammation in the host is brought on by the parasite’s decomposition. The microfilariae and Wolbachia bacteria (confined to hypodermal cells of lateral cords in microfilariae) coexist in a mutualistic symbiosis. These bacteria appear necessary for filarial growth and survival [6]. The development of inflammatory diseases is also greatly influenced by these endosymbionts; thus explaining the use of antibiotics like doxycycline (or azithromycin) which delivers a macrofilaricidal activity and modulates inflammatory processes [6,7]. Diethylcarbamazine, ivermectin, and albendazole were prescribed as medical treatments for microfilariae. Opoku et al. [7] compared the outcomes of 1500 individuals with *O. volvulus* skin infection that were treated with single doses of either of the two macrocyclic lactones, ivermectin (half-life 1 day) or moxidectin (half-life 1 month) [8]. After one year of therapy, the percentage of participants without detectable cutaneous microfilariae was 2.5-fold higher in the moxidectin group (38.4%) than in the ivermectin group (15%) [7,8]. The WHO advises ivermectin therapy at least twice a year for more than 10 years, but so far there is limited success in the African continent. Despite many rounds of sustainable ivermectin mass drug administration (Mectizan^®^, donated by Merck (Rahway, NJ, USA)) with 80% therapeutic coverage, onchocerciasis persists in adults in many African villages, with more time needed to reach elimination of the disease [9,10,11,12,13,14].

Onchocerciasis is the second-most common infectious cause of blindness in the world, causing vision loss in 1.15 million individuals [1]. Despite these epidemiologic statistics [15,16,17,18,19,20,21,22,23,24], little has been written about onchocerciasis in the recent retina and uveitis literatures [25,26,27,28,29,30,31,32,33,34,35,36,37,38,39,40,41,42,43,44,45,46,47,48,49,50,51,52,53,54,55,56]. This renders young ophthalmologists, especially in the developed world, unfamiliar with the clinical features. With increasing international travel or hunger and poverty driven mass global population immigration, clinicians in nonendemic countries must be familiar with imported neglected tropical diseases.

The risk of visual impairment increases, in part, with rising infection prevalence and intensity in a population [5]. In some villages, the prevalence of infection can range from 80 to 100 percent by age 20, with blindness reaching its peak between 40 and 50 years of age.

The two largest retina image banks totaling more than 66,000 multimodal images (Retina Image Bank from American Society of Retina Specialists and Retina Rocks Retina Image and Reference Library from Retina World Congress) had no photographs related to onchocerciasis. Due to tourism or migration in recent years, parasitic diseases, which cause major ocular morbidity in certain areas, have been moving from endemic areas to other regions of the world [10,44,45]. Our aim is to present a case series of chorioretinitis related to onchocerciasis with the diagnosis delayed due to its rarity in the developed world and its polymorphic appearance.

## 2. Materials and Methods

Case series of chorioretinitis cases seen in different parts of the developed world where diagnosis was missed for a long time. All subjects gave their informed consent for inclusion in this study. The study was conducted in accordance with the Declaration of Helsinki. National legislation indicated that ethics approval is not required for this type of retrospective anonymized noninterventional study. The methodology relied on serial fundus photograph, intravenous fluorescein angiography, and optical coherence tomography; all collected retrospectively from various centers.

## 3. Results

### 3.1. Case 1

A 46-year-old White Middle Eastern man who worked in Abidjan (Côte D’Ivoire, Africa) for the past 23 years. He developed painful red swelling of the eyes in 1998 one year after initial visits to the neighboring river. Ten years later (2008), he developed blurry vision bilaterally. Fluorescein angiography (FA) revealed disc staining that was interpreted as drusen of the optic nerve head. No retinal scars were detected except for peripapillary subretinal tracts (Figure 1, Figure 2 and Figure 3). Subsequently several specialists in Côte D’Ivoire diagnosed papillitis and five consecutive yearly magnetic resonance (MR) brain imaging scans were performed, and all were negative. In 2012 and 2013, multiple chorioretinal scars were evident at the equator with large area of peripapillary atrophy. In 2016, extensive uveitis workup (normal CBC-no eosinophilia; purified protein derivative (PPD) skin test; Venereal disease research laboratory (VDRL); Chest radiograph), failed to reveal positive tests except for toxoplasmosis (IgM 0.66 IU/mL-negative < 0.8; IgG 418 IU/mL-positive > 3). Intraocular dexamethasone implants were given to quiet the uveitis in the left eye with the working diagnosis of toxoplasma chorioretinitis. In June 2018, he presented with 20/40 vision right eye, and finger counting 20 cm left eye. He had a severe flare in the anterior chamber and dense nuclear sclerosis in the left eye. Our initial impression was river blindness (Figure 4 and Figure 5). He was given azithromycin, a dose of 1 tablet for 12 days to be followed by a single dose of moxidectin 8 mg. The patient was not able to procure moxidectin. A complete work-up was repeated in five different medical centers and is briefly summarized. Negative tests included PPD skin test; Quantiferon Gold for tuberculosis; chest X-ray; computed tomography (CT) chest scan; VDRL; rapid plasma reagin (RPR); TPHA Treponemal pallidum hemagglutination; complete blood count; erythrocyte sedimentation rate (ESR) (8 and 13 mm/hour); C-reactive protein (CRP); rheumatoid factor; HLA B27; p-antineutrophilic cytoplasmic antibody (ANCA) IgG; c-ANCA IgG; anti-nuclear antibody (ANA); C3 and C4 complement; anti cyclic citrullinated peptide (CCP); anti ds DNA; Brucella IgG and IgM (Elisa) and Wright’s agglutination tests; Widal serum agglutination H and O tests (*S. paratyphi* and *S. typhi*). A borderline test included fluorescent treponemal antibody absorption (FTA-ABS) IgM titer of 1/10 (negative < 1/10). Elevated titers were for angiotensin converting enzyme 85 U/L (Normal range 8–65 U/L) in addition to IgG toxoplasma as mentioned before. A consultant dermatologist found no skin lesion to biopsy. On 27 September 2019, the patient presented with an attack of uveitis in both eyes, and he responded to one dose of ivermectin (12 mg) and doxycycline 100 mg for 30 days with resolution of the uveitis 2 months after ivermectin. Following a one-year-course of quarterly ivermectin, he underwent cataract extraction on 15 July 2020 and was symptom free for 3 months. He failed to follow up since. The patient was contacted three years later; 25 years after his first eye symptoms. He reported having current and past recurrences of the uveitis. He refused further medical intervention. The serial fundus photographs, autofluorescence, fluorescein angiography and SD-OCT were collated and shown in chronological order over an 18-year period (Figure 1 and Figure 2). No laboratory in the four Middle East centers visited by the patient had the facilities to do one of the three PCR: *Onchocerca volvulus*–specific qPCR-O150, pan-filarial qPCR melt curve analysis, or O150-PCR enzyme-linked immunosorbent assay.

### 3.2. Case 2

A 71-year-old White Middle Eastern man who is the cousin of Case 1 and lived together in the same household in Africa. He also visited frequently the nearby river. He had no visual symptoms. Visual acuity was 20/20 in both eyes. Large deep chorioretinal scars were noted temporally with adjacent zone of RPE alterations in both eyes (Figure 6). Subsequently, similar blood workup to Case 1 was requested. The infectious disease team concentrated on more urgent issues. He had negative workup (HIV 1 and HIV antibody and p24 antigen, CMV PCR, EBV PCR, ANA, C3 complement, glomerular basement membrane antibody, hepatitis B PCR, ANCA, fecal parasite concentration, stool culture, stool Wright’s stain test, chest radiograph, total body magnetic resonance imaging) except for anemia (hematocrit 25), plasma cell myeloma (bone marrow biopsy), hyperuricemia, and rapidly progressive glomerulonephritis (systemic hypertension, proteinuria, increased C4 complement to 0.46 g/L). Because of the absence of eye symptoms, multiple system disease, and the inability of the facility to perform PCR onchocerciasis, the patient was not asked to come for an eye follow-up.

### 3.3. Case 3

A 67-year-old Black African woman was referred for decreased vision after cataract surgery. The best corrected visual acuity was 20/125 right eye and 20/50 left eye, with a posterior chamber intraocular implant in the right eye and dense nuclear sclerosis in the left eye. Posterior vitreous detachment was noted bilaterally without vitritis. The right fundus had crystalline deposits noted by a funduscopy and shown as hyperreflective mid-retinal foci using an SD-OCT (Figure 7). A diffuse, patchy loss of the ellipsoid in the posterior pole of the right eye (Figure 7). Supertemporal linear subretinal fibrosis tracks in the right eye (Figure 7) were initially interpreted as the spontaneous reattachment of a detached retina. The inferior retina revealed perivascular sheathing along the inferotemporal arcade and inferonasal perivascular pigmentary deposits (Figure 6). She reported a decrease in vision in the right eye since age 30. Systemic history was positive for hypertension well controlled on beta-blocker. The patient immigrated recently to Philadelphia from Sierra Leonne, where she lived by the riverside. She reported a heavy consumption of Kola nuts. Blood workup (including CBC, hemoglobin electrophoresis (for sickle cell disease), RPR, FTA-ABS, Quantiferon Gold for tuberculosis, angiotensin converting enzyme, and lysozyme) and chest radiograph were all negative. The preliminary differential diagnosis was uveitic crystalline maculopathy, West African crystalline retinopathy and resolved retinal detachment. This case was presented during one of the International Retina Rounds and one expert suggested the diagnosis of ocular onchocerciasis. The patient did not return for follow-up to do PCR tests and dermatologic consultation.

### 3.4. Case 4

A 49-year-old White Middle Eastern man who has lived for 3 decades near swamps connected to the Wouri River in Cameroon. Upon presentation, best corrected vision was 20/600 right eye and 20/80 left eye. Optic atrophy was significant bilaterally (Figure 8). Fine retinal tracts were noted in the left eye. Systemic and uveitis workup (including absence of eosinophilia, MR Brain, ANA, malaria, toxoplasma and syphilis serology) and coagulation screen (Factor V Leiden, VIII, antithrombin III) were negative. A tropical disease was suspected, and all documented ocular history was requested. A large juxtafoveal temporal patch of stippled retina had outer retinal thinning by OCT was discovered 14.5 years ago. Subsequently, 15 months ago he developed uveitis of the left eye and left papillophebitis with macular detachment that was well controlled by oral corticosteroids. Recorded 100 days later, the right eye had papillophlebitis with cystoid macular edema that was interpreted as central retinal vein occlusion. He received a one-month course of oral corticosteroid and one intravitreal injection of bevacizumab. Our working diagnosis was presumed ocular onchocerciasis as a PCR for *Onchocerca volvulus* was not accessible.

## 4. Discussion

Little is known about multimodal findings in the setting of river blindness and most reports detail short follow-up without fundus documentation. Over the past thirty years, most of the publications appeared in epidemiologic or infection or tropical disease literature with no fundus photograph familiarizing the disease in the ophthalmic or retina or uveitis literatures. We present a long-term follow-up of a case series where the clinical diagnosis was obscure and shows the progressive disease with a myriad of findings captured by multimodal imaging; presenting unique fundus findings that help in the diagnosis of the disease.

The ocular findings in onchocerciasis are very varied (Table 1). Our aim is to refresh these findings and add multimodal findings plus long-term photographic documentation of this progressive pleomorphic disease. All the presented cases had an indefinite diagnosis for a prolonged time. Diagnosing this disease is time costly and presents a difficult challenge to the ophthalmologist as most infected individuals manifested no disease symptoms. Without skin findings, the gold standard diagnostic test (i.e., the skin snip) could not be applied.

The road to diagnosis of onchocerciasis chorioretinitis: 1-search for travel or residency in endemic areas; 2-access past fundus photographs to assess progression; 3-exclude all known causes of uveitis or optic neuritis (complete work-up).

Onchocercal choroiditis is a condition that has been studied more than 3 decades ago experimentally [26,45], observationally [15], and epidemiologically [16,17]. First, after injecting 100,000 live *Onchocerca volvulus* microfilariae intravitreally, the retinal pigment epithelium gradually became atrophic in patches along with inflammation and thinning of the outer retina. Second, little was known about the evolution of choroidal lesions over the long term. It was Semba et al. [15] that tried in 57 patients to track the evolution of onchocercal chorioretinitis over a period of 1 to 3 years. In fresh cases, live retinal microfilariae, retinal hemorrhages, and fine retinal pigment epithelium (RPE) alterations are noted; subsequently, there is growth of the depigmentation zone (200 microns yearly) at the borders of the chorioretinal scars regardless of treatment. Third, the epidemiology of onchocerciasis in rain forest regions in Africa and South America provided a deep insight into the disease severity. This first epidemiological investigation involved 800 inhabitants of a rubber plantation amidst a hyperendemic region of the Liberian rain forest where 84% had the infection; of these, 29% had intraocular microfilariae, and 2.4% were blind in one or both eyes. All cases of bilateral blindness and one-third of visual impairment cases were caused by onchocerciasis. Chorioretinitis was detected in three-quarters of the participants and was the direct cause for half of the visual impairment. The presence of retinitis, subretinal fibrosis, and optic neuropathy was found to be strongly correlated with uveitis. In this second epidemiologic study from Ecuador’s rain forest onchocerciasis focus, Cooper et al. [33] examined 785 infected people all with a positive skin snip. Onchocerciasis caused 0.4% of blindness, 8.2% of visual impairment, 5.1% of optic atrophy, and 28.0% of chorioretinopathy. Early detection and treatment is of great importance as chorioretinitis becomes irreversible when treated late or inadequately [24] (Table 2).

Our case series establish, 1-peripapillary atrophy; 2-subretinal tracts showing trajectory of the microfilariae; temporal chorioretinal lesions of varying sizes [29]: dot size, coin size, round lesions, or torpedo lesions, or any form; chorioretinal atrophy adjacent to toxoplasma like deep chorioretinal scars; retinitis and vitritis attacks; optic neuritis attacks; optic atrophy; retinal vasculitis; retinal venous engorgement or central retinal vein occlusion-like symptoms accompanied by papillophlebitis. Serial exams allowed us to witness the changing face of onchocerciasis with follow-up beyond ten years documented photographically. FA demonstrated retinal vasculitis and disc leakage as well as cystoid macular edema. An OCT documented outer retinal atrophy around peripapillary choroiditis, disc edema in optic neuritis, crystalline deposits in macula and cystoid macular edema. Future OCT studies may further document microfilaria in the conjunctiva, cornea, anterior chamber, or retina in heavily infected eyes in mesoendemic regions during the acute phases. However, no such photographic documentation exists to date. OCT was instrumental in detecting disc swelling, especially in “normal-appearing” discs (Figure 5); similarly, visual fields testing can detect major defects in apparently normal discs [34]. Moreover, optic neuritis or atrophy accompanied by temporal retinal mottling or OCT signs of outer retinal atrophy appears to be characteristic of onchocerciasis [34].

More than half of the ocular fluids from individuals with ocular onchocerciasis contained autoimmune antibodies that were directed against the outer region of the photoreceptor and were unrelated to either the interphotoreceptor retinoid binding protein or the retinal S-antigen (S-Ag) [40,41]. These anti-retinal antibodies could contribute to the retinal degeneration brought on by onchocerciasis [40]. According to several investigations, posterior segment disorders may be exacerbated by cross-reactive antibodies produced in response to the antigens of O. volvulus (Ov39) and the retinal pigment epithelial (RPE) antigen (hr44). [18,19]. It is unclear if the persistence of microfilariae or their byproducts in the posterior segment or autoimmune reactions are to blame for these progressive ocular alterations.

The diagnosis of infection in a person has varied, with the classical clinical presentations varying from being obvious in heavily infected cases (e.g., observation of microfilaria in the cornea or anterior chamber, detecting the parasites in skin-snip biopsy, or finding a palpable subcutaneous adult worm) to being relatively insensitive in patients carrying lower loads of the parasite. The incubation period can be prolonged for up to 15 months. Onchodermatitis resembles eczema with varying degrees of papular, lichenoid, atrophic, and pigmentary alterations [50]. Eosinophilia is not a sensitive indicator of *Onchocerca volvulus*, according to recent clinical investigations, with one-third of patients having a normal eosinophil count. The currently recommended epidemiological diagnostic method entails the measurement of an antibody response to the parasite antigen Ov16 with 20% false negative results [46].

More recent research suggests that it is largely the Wolbachia bacteria (which are endosymbionts) which cause the immunogenic response [46]. It appears that there are two main strains of *O. volvulus*, the savanna strain that causes ocular disease, even with moderate parasite burdens, and the rainforest strain, that does not lead to blindness despite high parasite burdens. This predilection for ocular disease seems related to higher quantities of Wolbachia.

Ivermectin binds to the inhibitory neurotransmitter GABA on neurons and muscles leading to the activation of a chloride influx, hyperpolarization of the membrane, resulting in paralysis and death of microfilariae. The life span of microfilariae is 1–2 years while the clearance of microfilariae has been studied extensively [38,39]. Skin microfilariae are reduced by half in 24 h, 94% in one week, and 98% by 4 weeks following a single dose of ivermectin with the clearance from the anterior chamber lagging by several months. Microfilariae repopulate the eye several months after a single dose of ivermectin from the continuous production of microfilariae by the adult worm. This explains the WHO recommendation of biannual ivermectin to be administered over a span of 10 to 15 years [38]. Even children as early as 4-years should be included in the massive drug administration (dose 150 μg/kg) according to a recent meta-analysis [37,39]. Note that there is ample evidence that moxidectin appears to offer better control than ivermectin [38,39].

One major challenge facing elimination of *O. volvulus* transmission following mass anti-filarial therapy administration in sub-Saharan Africa is the very variable response of adult worms to the embryostatic effect of ivermectin as well as the variable number of adult worms per person [55]. Moreover, recent literature has suggested the appearance of strains of *O. volvulus* resistant to ivermectin. Therefore, other treatments have been tried with varying levels of success include azithromycin and rifampin. Opoku et al. [7] compared the 18 months’ results of single doses of ivermectin and moxidectin on around 1500 subjects infected with *O. volvulus* microfilariae. The proportion of subjects with undetectable skin microfilariae at one-year post-treatment was 38.4% in the moxidectin compared with only 15% in the ivermectin group [8]. This longer lasting effect of moxidectin related to the long half-life (moxidectin 20–43 days, ivermectin < 1 day) [8]. Ophthalmologists would like to extrapolate the results of the skin effect to the choroid in the hope of curing this cause of blindness. Cousens et al. [35] in his field study in mesoendemic communities in Nigeria concluded that annual delivery of ivermectin in a sustained fashion could halt onchocercal blindness from optic atrophy. These findings were confirmed by different investigators [18,36,48].

In Case 4 where venous impedance followed disc edema, treating the disc edema can lead to prompt return of normal retinal venous pattern. In the first eye, systemic corticosteroid and intravitreal anti-vascular endothelial growth factor achieved good control of the disc edema and cystoid macular edema, while intravitreal dexamethasone implant in the fellow eye achieved similar response. It is well known that when early central retinal vein occlusion or venous stasis is deemed secondary to optic nerve swelling or neuritis, corticosteroid whether systemic or local, alleviate optic nerve swelling, thereby relieving compartment obstruction at the level of the lamina cribrosa, and improving venous outflow [47].

Ocular lesions in river blindness are not specific including corneal scars, uveitis, cataract and chorioretinitis. Differential diagnosis of ocular onchocerciasis chorioretinitis [56,57,58,59,60,61,62] include toxoplasmosis, sarcoidosis, tuberculosis, larva migrans, syphilis, diffuse unilateral subacute neuroretinitis (DUSN) from various nematodes (*Toxocara canis*, *Ancylostoma caninum*, *Strongyloides stercoralis*, *Ascaris lumbricoides*, and *Baylisascaris procyonis)*, schistosomiasis [57], cestodes [58], and other microfilaria (including *Marsonella perstans*, Loa loa, *Onchocerca gutturosa*, or *Dracunculus medinensis*). Oculocutaneous tropical disease differential includes tuberculosis, leprosy, Chagas disease (also known as American trypanosomiasis), sporotrichosis (also known as “rose gardener’s disease”) [59], coccidioidomycosis (also known as San Joaquin Valley fever) [60], leishmaniasis [56], giardiasis [56] and infection by other microfilaria (*Onchocerca gutturosa*, *Loa loa*, *Dracunculus medinensis*, *Marsonella perstans*). In our cases, the most specific differential diagnoses included ocular toxoplasmosis and DUSN [61,62]. Onchocerciasis and DUSN share common features such as subretinal tracks, focal RPE changes, and small, white subretinal spots. Differentiating features include laterality, difference in fundus changes, and different zones of endemicity. Classically, DUSN occurs unilaterally. In late DUSN, with late diffuse RPE degeneration, retinal arteriolar narrowing and optic atrophy. There are specific endemic areas for DUSN where causative parasites reside along with the carrier animals of etiologic worms (Caribbean islands, Brazil, Venezuela, India, China, Midwest USA). Our cases lived in endemic areas in Africa for onchocerciasis, had bilateral involvement, and did not have the wipe-out syndrome in late DUSN, and had a relatively preserved visual acuity. Ocular toxoplasmosis manifests as fluffy white focal necrotizing retinitis adjacent to a pigmented chorioretinal scar. This is often a punched-out scar in an immune competent individual, while it has a fulminant course in immunocompromised subjects. Unusual presentations include diffuse or punctate outer retinitis and occlusive vasculitis. In the current cases, new lesions arose distant from previous foci, subretinal tracts, subretinal dots, and intraretinal, round, coin-like lesions without surrounding inflammation are not seen in ocular toxoplasmosis. Control of uveitis upon initiation of ivermectin favors the diagnosis of presumed ocular onchocerciasis.

The drawback of this paper is absence of superficial skin lesions that can offer a chance for snip biopsies as well as a lack of adjuvant tests like PCR and serology. Historical evidence (living near rivers in endemic areas), negative workup for other etiologies, long-term follow-up, characteristic clinical findings and resolution of uveitis following ivermectin therapy are the five cornerstones for the current presumed diagnosis.

## 5. Conclusions

Live microfilariae caused little or no inflammation. The immune reaction to degenerating dead worms resulted in intraocular inflammation (uveitis, keratitis, chorioretinitis, optic neuritis) with progressive visual loss. Peripapillary chorioretinitis, retinal tracks, papillophlebitis, retinal vasculitis, and polymorphic chorioretinitis characterize the disease. Experimental and clinical data indicated a cardinal role of the endosymbiotic Wolbachia bacteria in the pathogenesis of blindness. In addition to ivermectin therapy, oral administration of doxycycline resulted in significant decrease in Wolbachia bacterial load, altered filarial embryogenesis and ultimately a reduced ocular inflammatory response.

Ocular onchocerciasis should be suspected in subjects residing near sub-Saharan rivers, subjects with disfiguring skin disease or seizures. With the recent influx of African immigrants, ocular onchocerciasis may be seen more often in clinics in temperate climates in immigrants and travelers from endemic regions. Familiarity of ophthalmologists with the various ocular signs corneal scar, evolving chorioretinitis, optic neuropathy would help recognize the disease. Blindness caused by *O. volvulus* results in significant morbidity, long-term disability, reduced economic productivity and life expectancy. Clinicians should have a high suspicion for imported onchocerciasis which is in the rise among people residing in endemic areas with vision changes, subcutaneous nodules, or itchy skin rashes. The parasite that gained historical notoriety as “African river blindness” needs to be revived in our retina and uveitis clinics.

## Figures and Tables

**Figure 1 diagnostics-13-03626-f001:**
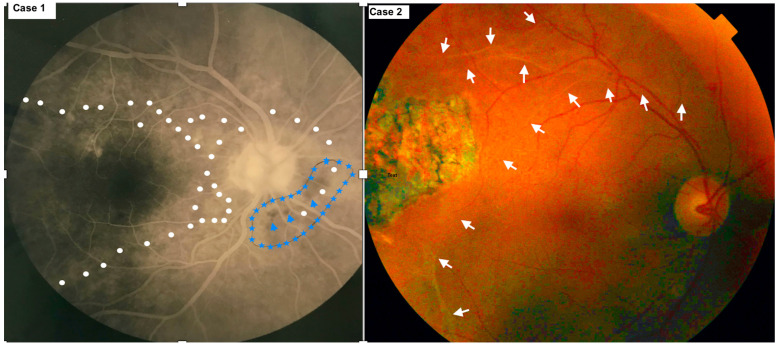
Subretinal tracts of microfilaria of right eye in Case 1 ((**left**) side of montage) (white dots and blue stars on fluorescein angiography (FA)) and right eye of Case 2 ((**right**) side of montage) (white arrows on fundus photography).

**Figure 2 diagnostics-13-03626-f002:**
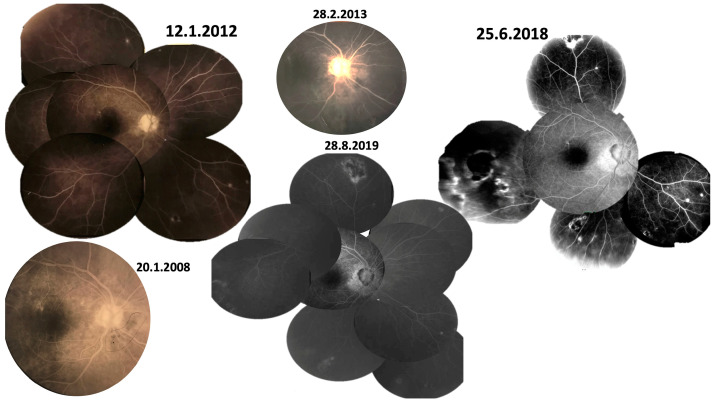
(Case 1; right FA). Taken after a 12-year follow-up showing the new onset of peripapillary atrophy.

**Figure 3 diagnostics-13-03626-f003:**
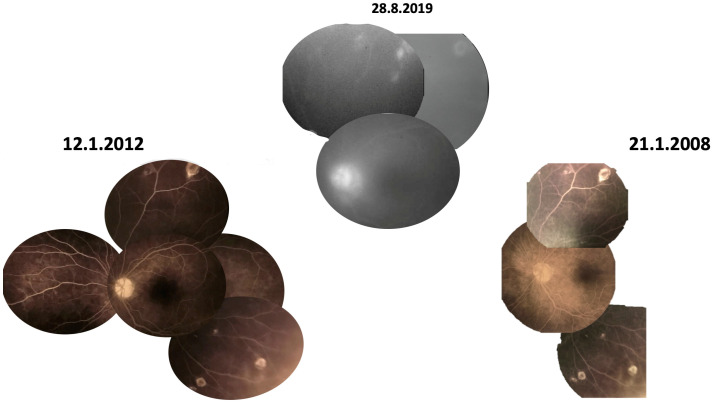
(Case 1; left FA). Taken after a 12-year follow-up showing active vitritis with optic nerve and peripapillary inflammation with one new focus of retinitis superiorly. Torpedo or round lesions are noted at the equator with central blocking fluorescence and a surround with fluorescein staining.

**Figure 4 diagnostics-13-03626-f004:**
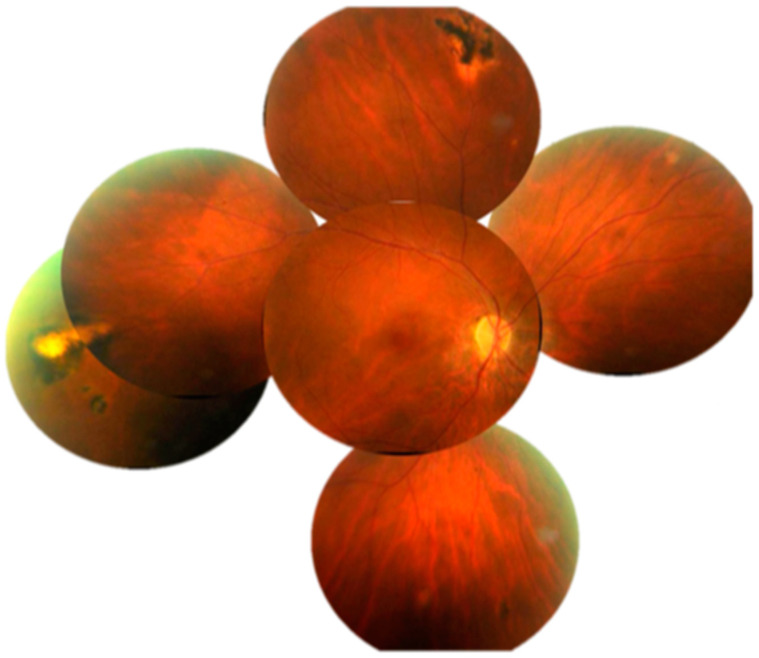
(Case 1). Montage of the right fundus on 7 October 2020. Noted are the following: Optic pallor, one disc diameter of peripapillary atrophy with pigment stipling; retinal arterial attenuation; small yellowish pinpoint or coin-shaped lesions at the equator; multifocal pigment-centered lesions with yellow surround or yellow-centered lesions with pigment surround.

**Figure 5 diagnostics-13-03626-f005:**
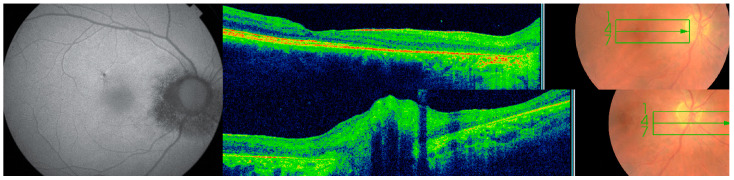
OCT macula, optic disc, and fundus autofluorescence on 25 June 2028. There is attenuation of the outer nuclear layer and disruption of the ellipsoid zone extending from nasal macula to the peripapillary area. Subclinical disc swelling is well noted by SD-OCT. Peripapillary ring of hypo-autofluorescence is characteristic of onchocercal chorioretinitis.

**Figure 6 diagnostics-13-03626-f006:**
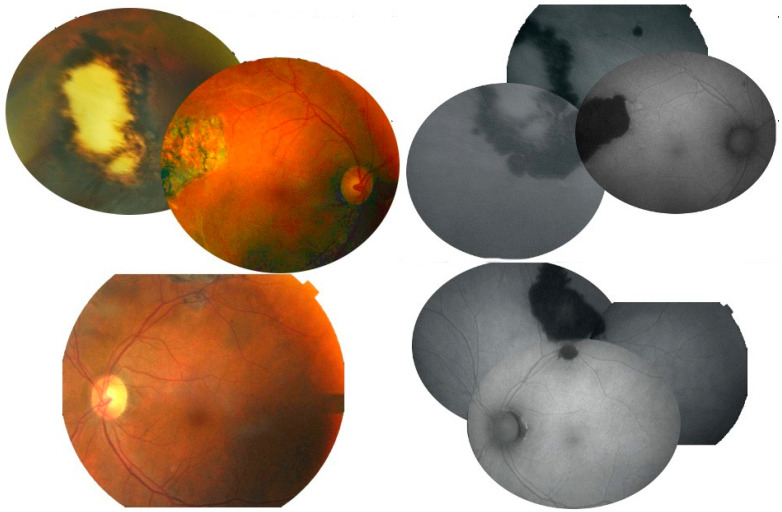
(Case 2). Montage of fundus photograph and fundus autofluorescence of both eyes show small lesion beside large geographic lesions temporal to the fovea right eye and superior to the fovea left eye.

**Figure 7 diagnostics-13-03626-f007:**
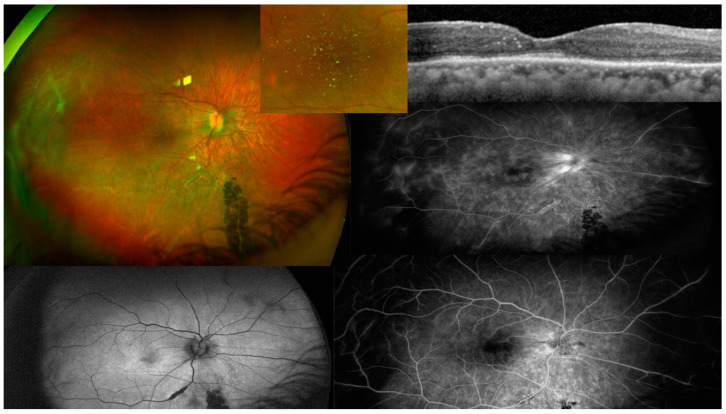
(Case 3). The right eye had microfilaria tract superotemporally, crystalline deposit in fovea, patchy ellipsoid zone disruption, vasculitis with perivascular pigmentary deposit.

**Figure 8 diagnostics-13-03626-f008:**
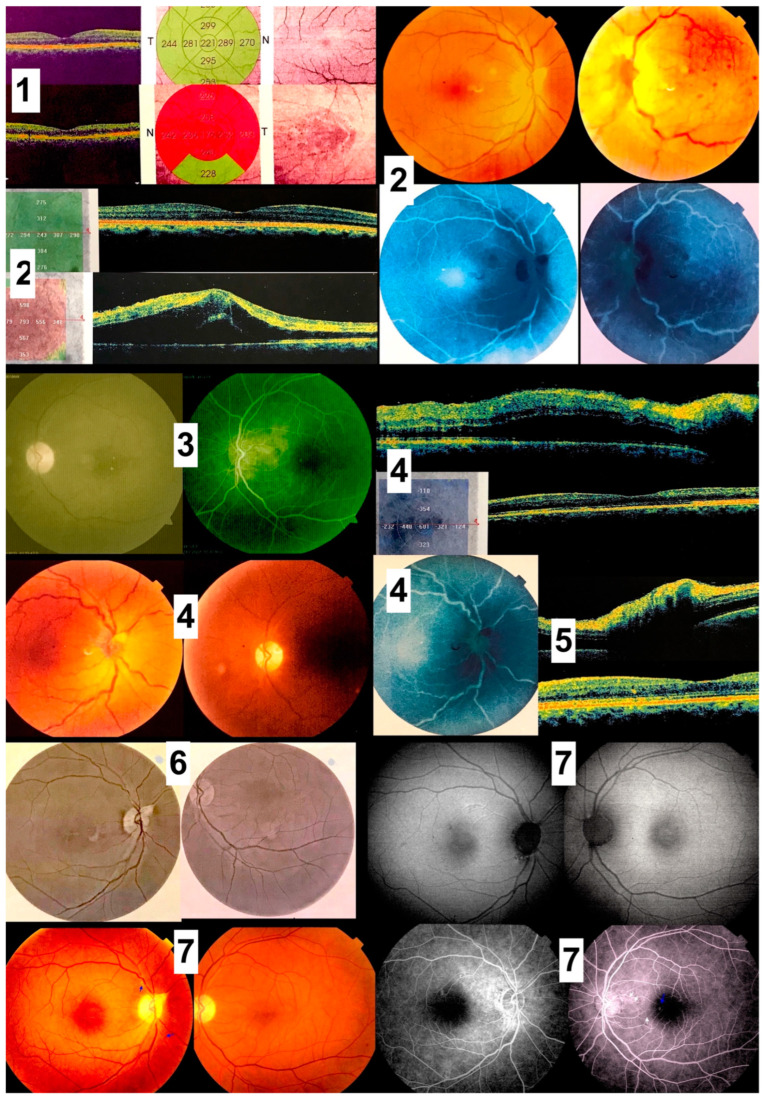
(Case 4). A composite of fundus photographs, FA, auto fluorescence and OCT. 1—1 January 2009 initial OCT scan: outer retinal thinning left macula with pigment stippling; 2—24 May 2022 left papillophlebitis 13.5 years after initial presentation; 3—17 July 2022 left papillophlebitis resolved after oral corticosteroids; 4—9 September 2022 right papillophlebitis 100 days after left papillophlebitis; 5—19 October 2022 disc edema right eye; 6—29 December 2022 resolved papillophlebitis after intravitreal bevacizumab and oral corticosteroids. Peripapillary fibrosis is noted superiorly; 7—22 May 2023 the last follow-up 14.5 years after presentation revealed bilateral optic atrophy and subtle subretinal tracts (arrows). Peripapillary fibrosis is noted.

**Table 1 diagnostics-13-03626-t001:** Ocular findings in onchocerciasis.

Site	Findings	Frequency	References
Adnexa	Adnexal nodule	Rare	[32]
Conjunctiva	conjunctivitis	common	
	phlyctenule	common	
	microfilaria	common	
Cornea	motile microfilaria	13.8–95.7%	[7,29,31]
	punctate keratitis (snowflake)	13.80%	[16,31]
	sclerosing keratitis	5%	[16]
	neovascularization	rare	
Sclera	microfilaria	rare	[27]
Anterior chamber	motile microfilaria	13.6–40%	[7,16]
	glaucoma	1.60%	[16]
	iridocyclitis	8%	[31]
Lens	Cataract	infrequent	
Vitreous	microfilariae	infrequent	[27]
Optic disc	optic neuritis	1–5.5%	[16]
	optic atrophy	5.1–57%	[16,28,35]
	microfilariae	rare	[28]
	epipapillary fibrosis	uncommon	[29]
Retina-choroid	chorioretinitis	38–75%	[15,29]
	peripapillary atrophy	9–25%	[28,29]
	mottled fundus	common	[29]
	retinal vasculitis	1%	[16]
	retina tracts	10%	[16]
	white intraretinal deposit	21.40%	[16]
	microfilaria	rare	[27]

N.B. Results vary according to cross sectional vs. longitudinal design, degree of fly infestation, length of follow-up, study design (field vs. clinic exam), age and gender of participants.

**Table 2 diagnostics-13-03626-t002:** Comparison between experimental [26,45] and human onchocerciasis [15,46].

Model	Monkey	Human
Uveitis	Common	Common
RPE changes	Common	Common
Disc edema/Atrophy	Common	Common
Retinal vasculitis	Common	Infrequent
Venous engorgement	Frequent	Uncommon
Retinal hemorrhages	Common	Uncommon
Histopathology	Eosinophilic choroiditis	Eosinophilic choroiditis [27,28]

## Data Availability

Data is contained within the article.

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
