# Peer review of "Presumed Onchocerciasis Chorioretinitis Spilling over into North America, Europe and Middle East"

_diagnostics, 2023, doi:10.3390/diagnostics13243626_

Round 1
Reviewer 1 Report (Previous Reviewer 2)
Comments and Suggestions for Authors
This revised version of the paper by Mansour et al on presumed onchocerca chorioretinitis is significantly improved particularly with reference to the use of ivermectin.
Author Response
We are grateful for the positive remarks of our esteemed reviewer
Reviewer 2 Report (New Reviewer)
Comments and Suggestions for Authors
Many thank to the Editor for an opportunity to review this manuscript. In their paper Dr Mansour et al reports four cases of ocular onchocercosis. Although the topic of the paper is interesting, the quality of illustrations is not very high. Moreover, the paper contains many imprecisions and typos. Some noted below. Also I cannot agree with the title “Commonly Missed Diagnosis Spilling Over into North America, Europe and Middle East” since the authors did not analyze diagnosing of onchocercosis in those regions and cannot say how often it is missed. It is rather a suggestion.
Please add minimal details on the methods in the abstract.
27 “Typical skin findings are often absent in eyes” pleaserephrase
30 “Onchocercid” ?
56 “The microfilariae and Wolbachia bacteria coexist” how exactly they do
73 please remove abbreviation
87-89 duplicates 74-76
91-94 it is interesting note please put in before saying the aim of the study
Methods: I understand difficulties in describing methodology od such study but you have at least to try note which kind of imaging data were collected.
119-133 and through the entire manuscript please spell out abbreviations for their first mention
137 collated
163 “Disorganization of nasal macula with atrophy of superficial retina and dissolution of the outer peripapillary retina” Please rephrase. This description sounds strange probably retina or OCT specialist can help. There are no such terms as “superficial retina” and “dissolution of the retina”
196 not clear why she had “nuclear sclerosis” after cataract surgery
Figure 9 is of poor quality I concern if it may be helpful to potential readers. Please provide better images
It is chorioretinitis as you and the referencing papers say
Please put section “discussion”
301-305 you also mentioned anterior uveitis…
338 please provide the explanation for
Author Response
Reviewer 2
Many thank to the Editor for an opportunity to review this manuscript. In their paper Dr Mansour et al reports four cases of ocular onchocercosis. Although the topic of the paper is interesting, the quality of illustrations is not very high. Moreover, the paper contains many imprecisions and typos. Some noted below. Also I cannot agree with the title “Commonly Missed Diagnosis Spilling Over into North America, Europe and Middle East” since the authors did not analyze diagnosing of onchocercosis in those regions and cannot say how often it is missed. It is rather a suggestion.
Reply:
The reviewers’ comments are very precise and aim to correct deficiencies in the submission. Many thanks for these instructive remarks and for the exhaustive time spent to review.
A higher quality montage for fig.9 was submitted.
We deleted commonly missed diagnosis as follows in the title:
Presumed Onchocerciasis Chorioretinitis: Commonly Missed Diagnosis Spilling Over into North America, Europe and Middle East
Changed to:
Presumed Onchocerciasis Chorioretinitis Spilling Over into North America, Europe and Middle East
Please add minimal details on the methods in the abstract.
Reply:
Methods: Multicenter case series of chorioretinitis or optic neuritis with obscure etiology that had serial multimodal imaging;
27 “Typical skin findings are often absent in eyes” please rephrase
Reply: Typical skin findings are often absent in such patients with chorioretinitis rendering the diagnosis more challenging
30 “Onchocercid” ?
Reply: Onchocercal chorioretinitis (some authors wrote Onchocercidae chorioretinitis and a couple wrote Onchocercid chorioretinitis. But the most common is onchocercal chorioretinitis)
56 “The microfilariae and Wolbachia bacteria coexist” how exactly they do
Reply:
The microfilariae and Wolbachia bacteria (confined to hypodermal cells of lateral cords in microfilariae) coexist in a mutualistic symbiosis.
73 please remove abbreviation
Reply: abbreviation not detected
87-89 duplicates 74-76
REPLY
The risk of visual impairment increases, in part, with rising infection prevalence and intensity in a population [5]. Partially correlated with rising infection prevalence and intensity in a community, visual impairment chances increased [5].
Changed to
The risk of visual impairment increases, in part, with rising infection prevalence and intensity in a population [5].
In some villages, the prevalence of infection can range from 80 to 100 percent by age 20, with blindness reaching its peak between 40 and 50 years of age. Before the implementation of control measures, hyperendemic areas were commonly abandoned due to high incidence of blindness. WHO programmes shifted from control to elimination in the American continent. WHO advises ivermectin therapy at least twice a year for more than 10 years, but so far there is limited success in the African continent.
Was changed to:
The risk of visual impairment increases, in part, with rising infection prevalence and intensity in a population [5]. In some villages, the prevalence of infection can range from 80 to 100 percent by age 20, with blindness reaching its peak between 40 and 50 years of age.
91-94 it is interesting note please put in before saying the aim of the study
Reply:
Due to tourism or migration in recent years, parasitic diseases, which cause major ocular morbidity in certain areas, have been moving from endemic areas to other regions of the world [10,44,45]. Our aim is to present a case series of chorioretinitis related to onchocerciasis with the diagnosis delayed due to its rarity in the developed world and its polymorphic appearance. The two largest retina image banks totaling more than 66,000 multimodal images (Retina Image Bank from American Society of Retina Specialists and Retina Rocks Retina Image and Reference Library from Retina World Congress) had no photographs related to onchocerciasis.
Was changed to
The two largest retina image banks totaling more than 66,000 multimodal images (Retina Image Bank from American Society of Retina Specialists and Retina Rocks Retina Image and Reference Library from Retina World Congress) had no photographs related to onchocerciasis. Due to tourism or migration in recent years, parasitic diseases, which cause major ocular morbidity in certain areas, have been moving from endemic areas to other regions of the world [10,44,45]. Our aim is to present a case series of chorioretinitis related to onchocerciasis with the diagnosis delayed due to its rarity in the developed world and its polymorphic appearance.
Methods: I understand difficulties in describing methodology od such study but you have at least to try note which kind of imaging data were collected.
Reply: This was added
Methodology relied on serial fundus photograph, intravenous fluorescein angiography, and optical coherence tomography, all collected retrospectively from various centers.
119-133 and through the entire manuscript please spell out abbreviations for their first mention
Reply
Subsequently several specialists in Côte D’Ivoire diagnosed papillitis and 5 consecutive yearly magnetic resonance(MR) brain imaging were performed, and all were negative. In 2012 and 2013, multiple chorioretinal scars were evident at the equator with large area of peripapillary atrophy. In 2016, extensive uveitis workup (normal CBC-no eosinophilia; purified protein derivative (PPD) skin test; Venereal disease research laboratory (VDRL); Chest radiograph), failed to reveal positive tests except for toxoplasmosis (IgM 0.66IU/ml- negative <0.8; IgG 418 IU/ml- positive>3). Intraocular dexamethasone implants were given to quiet the uveitis in the left eye with the working diagnosis of toxoplasma chorioretinitis. In June 2018, he presented with 20/40 vision right eye and finger counting 20cm left eye. He had severe flare in the anterior chamber and dense nuclear sclerosis in the left eye. Our initial impression was river blindness (Figures 4, 5). He was given azithromycin 1 tablet for 12 days to be followed by a single dose of moxidectin 8mg. The patient was not able to procure moxidectin. Complete work-up was repeated in 5 different medical centers and is briefly summarized. Negative tests included PPD skin test; Quantiferon Gold for tuberculosis; Chest X ray; computed tomography (CT) chest; VDRL; rapid plasma reagin (RPR); TPHA Treponemal pallidum hemagglutination; complete blood count; erythrocyte sedimentation rate (ESR) (8 and 13mm/hour); C-reactive protein (CRP); Rheumatoid Factor; HLA B27; p-ANCA IgG; c-ANCA IgG; ANA; C3 and C4 complement; anti cyclic citrullinated peptide (CCP); anti ds DNA; Brucella IgG and IgM (Elisa) and Wright agglutination tests; Widal serum agglutination H and O tests (S. Paratyphi and S. Typhi).
137 collated
Reply:
and 13mm/hour); CRP; Rheumatoid Factor; HLA B27; p-ANCA IgG; c-ANCA
changed to
and 13mm/hour); C-reactive protein (CRP); Rheumatoid Factor; HLA B27; p-antineutrophilic cytoplasmic antibody(ANCA) IgG; c-ANCA IgG; anti-nuclear antibody (ANA);
163 “Disorganization of nasal macula with atrophy of superficial retina and dissolution of the outer peripapillary retina” Please rephrase. This description sounds strange probably retina or OCT specialist can help. There are no such terms as “superficial retina” and “dissolution of the retina”
Reply:
Disorganization of nasal macula with atrophy of superficial retina and dissolution of the outer peripapillary retina.
Changed to
There is attenuation of the outer nuclear layer and disruption of the ellipsoid zone extending from nasal macula to the peripapillary area.
196 not clear why she had “nuclear sclerosis” after cataract surgery
Reply:
Case 3. This 67-year-old Black African woman was referred for decreased vision after cataract surgery. Best corrected visual acuity was 20/125 right eye and 20/50 left eye, with posterior chamber intraocular implant in the right eye and dense nuclear sclerosis in the left eye.
Figure 9 is of poor quality I concern if it may be helpful to potential readers. Please provide better images
Reply:
A higher quality montage was submitted for fig 9.
It is chorioretinitis as you and the referencing papers say
Reply:
Yes
Please put section “discussion”
Reply: The section was added
Discussion
301-305 you also mentioned anterior uveitis…
Reply:
15 lines further below, we have mentioned: “The presence of retinitis, subretinal fibrosis, and optic neuropathy was found to be strongly correlated with uveitis”.
338 please provide the explanation for
Reply:
OCT was instrumental in detecting disc swelling, especially in “normal-appearing” discs (fig.5); similarly, visual fields testing can detect major defects in apparently normal discs [34].
Round 2
Reviewer 2 Report (New Reviewer)
Comments and Suggestions for Authors
Thanks to the authors. They have adequately addressed my comments.
This manuscript is a resubmission of an earlier submission. The following is a list of the peer review reports and author responses from that submission.
Round 1
Reviewer 1 Report
Comments and Suggestions for Authors
This is an interesting retrospective case series of four patients with Onchocerciasis chorioretinitis.
Comments:
1. In lines 45-46, "many districts" was repeated twice.
2. In case 1, one could argue that the temporal lesion in the right eye was a toxoplasmosis scar.
3. You should state in each case presented the exact serology that was done. Also, you should state in case presentations and not at the end of the discussion that serological evaluation of Onchocerciasis or PCR to prove the infection was not done and that no diagnostics of the skin lesions could not be done.
4. Did case 2 do a Toxoplasmosis serology and what were the results?
5. Wherever you used the term "chemotherapy", the term "antimicrobial agent" or "antiparasitic medication" should be used.
6. In the article title and abstract you should state presumed Onchocerciasis chorioretinitis since you have only indirect clues for diagnosis.
Comments on the Quality of English LanguageOnly minor changes are needed.
Reviewer 2 Report
Comments and Suggestions for Authors
General comments
The paper presents a case series of ocular presumed microfilarial chorioretinitis. The authors have documented interesting clinical histories describing the clinical signs as they develop over time. The diagnosis of onchocerca chorioretinitisnis is based on "historical evidence (living near rivers in endemic areas), negative workup for other etiologies, long-term follow up and characteristic clinical findings are the four cornerstones for the current diagnosis". These are insufficient. Evidence of microfilarial disease elsewhere such as in the skin would be helpful to support the authors contention. There is a similarity to other microfilarial disease causing conditions such as DUSN.
Specific comments
What is the significance of fast flowing rivers?
Are the rivers faster?
Diagnosis could be supported by evidence of response to treatment with ivermectin
Comments on the Quality of English LanguageNone